# Stripping Model for Short Gamma-Ray Bursts in Neutron Star Mergers

Sergei Blinnikov [1,2,*] , Andrey Yudin [1], Nikita Kramarev [1,2] and Marat Potashov [1,3]

1   National Research Center "Kurchatov Institute", pl. Kurchatova 1, 123182 Moscow, Russia;
    yudin@itep.ru (A.Y.); kramarev-nikita@mail.ru (N.K.); marat.potashov@gmail.com (M.P.)
2   Sternberg Astronomical Institute, Lomonosov Moscow State University, Universitetsky pr. 13,
    119234 Moscow, Russia
3   Keldysh Institute of Applied Mathematics RAS, 4 Miusskaya Square, 125047 Moscow, Russia
*   Correspondence: sblinnikov@bk.ru

**Abstract:** We overview the current status of the stripping model for short gamma-ray bursts. After the historical joint detection of the gravitational wave event GW170817 and the accompanying gamma-ray burst GRB170817A, the relation between short gamma-ray bursts and neutron star mergers has been reliably confirmed. Many properties of GRB170817A, which turned out to be peculiar in comparison with other short gamma-ray bursts, are naturally explained in the stripping model, suggested by one of us in 1984. We point out the role of late Dmitriy Nadyozhin (1937–2020) in predicting the GRB and kilonova properties in 1990. We also review the problems to be solved in the context of this model.

**Keywords:** gamma-ray bursts; neutron stars; kilonova





## 1. Introduction

Long gamma-ray bursts (GRBs) are most likely produced in collapsing "hypernovae", a subgroup of type Ic supernovae (SNe) [1–6]. These events can be explained by the prompt collapse to a black hole (BH) of the core of a massive star ($M{\sim}40M_\odot$) that lost its outer hydrogen and helium envelopes. These SNe are called Hypernovae (HNe) since they usually have energies much larger than those of typical ones. Something essentially different was suggested for short GRBs well before the models of long ones. First ideas on neutron star + black hole (NS+BH) mergers related to short GRBs were put forward by Lattimer and Schramm ([7,8], also r-process was discussed there). A detailed scenario for NS+NS coalescence was worked out by Clark and Eardley [9], but no words on GRBs were mentioned in their important paper. They discussed in detail not just merging, but also **stripping** of a low mass neutron star (LMNS).

What is the difference between those mechanisms (merging vs stripping)? Let us take a look at the sketch in Figure 1. On the left there are two neutron stars orbiting each other and approaching due to the emission of gravitational waves. The merging mechanism (the upper arrow) is as follows: becoming close enough, neutron stars literally merge into a single object in the last few revolutions. The single object is a black hole (as in the figure) or, less likely, a rapidly rotating neutron star. During the merging process, a part of the matter can be ejected from the system due to the tidal interaction [10]. In addition, due to the extreme acceleration of the matter rotation during the merger and its heating, an extremely collimated ejection of matter can form—a jet. This picture of what happens during the NS merger is generally accepted currently, e.g., [11], although many details are uncertain. The formation of jet is a complex and highly debated problem which goes beyond the main scope of our brief review. The main jet launching mechanism candidates are, in presence of a black hole remnant and a non-negiglible accretion disk, the Blandford–Znajek process [12] (see, e.g., [13] for a recent discussion of the role of magnetic field

geometry) and the neutrino–antineutrino annihilation process ([14,15], but see, e.g., [16] for results that disfavor this as the main jet-powering process in binary neutron star mergers). The jet may be mediated by neutrino annihilation or by magnetohydrodynamic processes [17] in the case when the remnant is a rapidly rotating, highly magnetized proto-neutron star [18–21], but how this is realistic is uncertain, see, e.g., [22–24] for recent contrasting results.

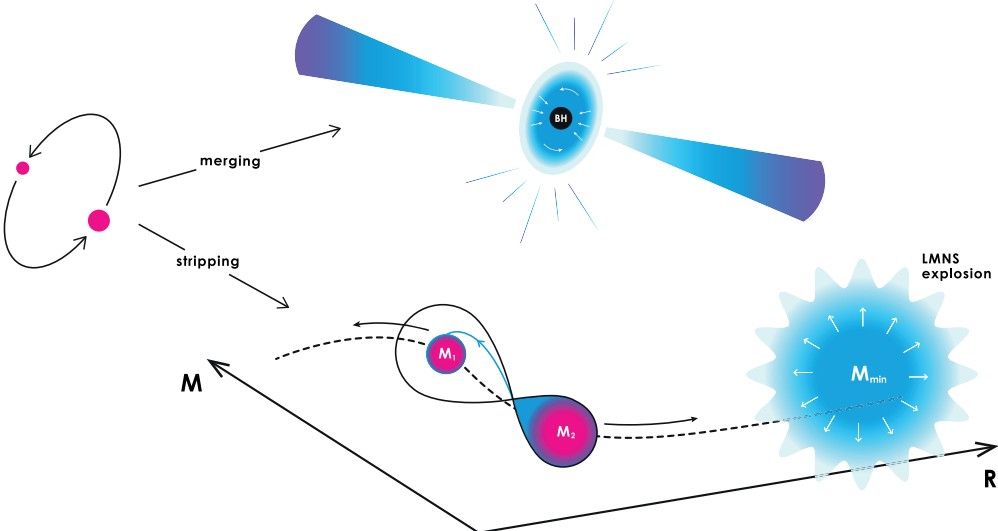

**Figure 1.** A sketch of the mechanisms of merging (arrow up) and stripping (arrow down). See text for details.

What is the essence of the stripping mechanism? Let us look again at the Figure 1. Note that this scenario requires sufficient asymmetry of the binary system component masses. Leaving for now the question of the exact meaning of the boundary between scenarios, let us follow the lower arrow. When the neutron stars are approaching each other, the component that has a smaller mass ($M_2$ in the figure) first fills its Roche lobe and begins to flow onto its more massive companion ($M_1$) through the inner Lagrange point. At the same time, the asymmetry of the system grows, the companions recede each other due to the approximate conservation of the system angular momentum. However, the flow does not stop (is stable) for two reasons: firstly, the loss of angular momentum for the emission of gravitational waves still drives neutron stars to each other, and secondly, the radius of a neutron star with a smaller mass begins to grow. The latter fact is implied by the properties of the NS mass-radius ($M-R$) relation, also shown in the Figure 1 by a dashed line. As one can see, $M_1$ grows, as a result of the mass exchange, and its radius remains approximately constant or decreases, and $M_2$ decreases as the radius grows. When $M_2$ reaches the value corresponding to the minimum possible NS mass ($M_{\min} \sim 0.1 M_\odot$), there will be an explosion, which, in fact, produces a gamma-ray burst.

This scenario was described in detail in the visionary paper [9], however, without mentioning the gamma-ray burst accompanying the LMNS explosion. The first firm prediction of GRB production in a binary neutron star coalescence was completed by Blinnikov et al. [25], and in the subsequent work [26]. D.K. Nadyozhin has carried out the hydrodynamic modeling of the explosive destruction of the LMNS. Many properties of the accompanying short GRB have been predicted in that important paper [26]. Among other things, it was shown there that the entire explosion process takes about a tenth of a second, while gamma radiation is generated by an outer layer of the neutron star matter, having very small mass, which is accelerated due to the cumulation effect up to semi-relativistic velocities.

However, this stripping scenario was forgotten for many years: first, due to the low energy of the resulting GRB, and secondly, due to the lack of a mechanism for generating an accompanying jet (the LMNS explosion is practically spherically symmetrical, see [27]).

In addition, there were serious doubts about the existence of a steady flow regime of matter during the stripping [28]. Everything changed after 17 August 2017.

## 2. GRB170817A and the Stripping Model

On 17 August, the gravitational wave detectors LIGO and Virgo caught the 6th GW-signal [29]. However, unlike the previous ones, this signal corresponded to the masses of merging objects, characteristic of NSs, not of black holes. After 1.7 s after the peak of the GW signal (remember this number) FERMI and INTEGRAL satellites detected GRB170817A [30], and 11 h later the source of the accompanying so-called kilonova [31] was discovered in the galaxy NGC4993, 40 Mpc away from us.

This impressive success of multimessenger astronomy was slightly overshadowed by the fact that this gamma-ray burst turned out to be rather peculiar. In particular, it was 10,000 times fainter than other known short gamma-ray bursts [30] and initially showed indistinct signs of a structured jet [32]. More recent VLBI observations explained it within an off-axis structured jet model [33]. Another model, that of the choked jet, was advanced in [34] by some authors of [33]. Although paper [34] predates [33], the citation and discussion of [33] is in footnote 6 of [34], where the latter write that "choked jet seems to be incompatible with GW1708178". Nevertheless, X-ray observations do not exclude that afterglow emission arises from a quasi-spherical mildly relativistic outflow [35,36]. Moreover, recent data confirm "a growing tension between the observations and the jet model" [37]. However, as we have shown in [38], many features of GRB170817A obtain their natural explanation in the short GRB stripping mechanism. Let us trace the sequence of events following [39] and Figure 2.

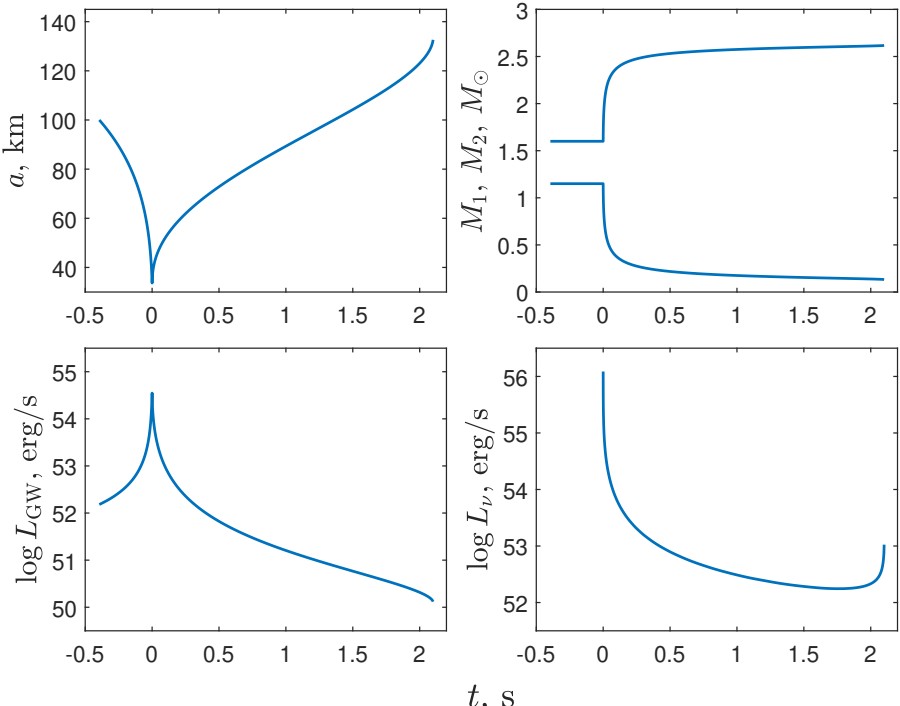

**Figure 2.** The distance between the components $a$, their masses $M_1$ and $M_2$, gravitational-wave $L_{\text{GW}}$ and "neutrino" $L_\nu$ luminosities as functions of time $t$ are shown.

So, two NSs with masses $M_1 = 1.6 M_\odot$ and $M_2 = 1.15 M_\odot$ approach each other due to gravitational radiation losses, with luminosity $L_{\text{GW}}$ (lower left panel) given by [40]

$$L_{\text{GW}} = \frac{32}{5} \frac{G^4}{c^5} \frac{M_1^2 M_2^2 (M_1 + M_2)}{a^5}. \tag{1}$$

when the distance *a* between NSs decreases to about 40 km (upper left panel), the less massive component $M_2$ fills its Roche lobe and accretion onto $M_1$ starts (top right panel). The components recede from each other. This recession, together with an increase in the asymmetry of the masses of the system, leads, according to the Formula (1), to a sharp drop in the luminosity of GW radiation. On the other hand, the matter of $M_2$ accreting onto the surface of $M_1$ leads to the energy release $L_\nu$ (lower right panel), which we estimate by the formula

$$L_\nu = \frac{GM_1\dot{M}_1}{R_1},\tag{2}$$

where $R_1$ is the radius of the first NS. Here, following [9], we attribute this energy release to the neutrino channel, although part of it will inevitably go out in the form of electromagnetic radiation. This radiation, by the way, can serve as a precursor before the gamma-ray burst itself on a characteristic time scale of the order of 1 s before the GRB (see, for example, [41]).

When the mass of $M_2$ is reduced to about $0.2M_\odot$, the stability of the overflow is lost (see the criterion Equation (4) below) and the star instantaneously, on the hydrodynamic time scale, reaches the minimum value of the NS mass $M_{\min}$ and explodes, producing a gamma-ray burst.

Thus, the most important parameter $t_{\mathrm{str}}$ appears in the stripping mechanism—the time interval between the peak of the GW signal and the gamma-ray burst. In our example, this parameter is slightly larger than 2 s. Its value is determined by the EoS BSk26 [42] used by us and the NS mass values, consistent with the LIGO-Virgo data on the progenitors of the GW170817 signal [43]. Strikingly, in the example discussed in [9], with mass values $M_1 = 1.3M_\odot$, $M_2 = 0.8M_\odot$ and legacy EoS, the authors nevertheless obtained exactly 1.7 s for $t_{\mathrm{str}}$!

*Boundary between the Two Scenarios*

The above example is, of course, only an illustration of what might be expected in a stripping mechanism. What will happen for other values of the component masses, all other things being equal? For an answer, let us turn to the Figure 3, which shows the value of $t_{\mathrm{str}}$ as a function of $M_1$ and $M_2$, for the area where the stripping mechanism "works".

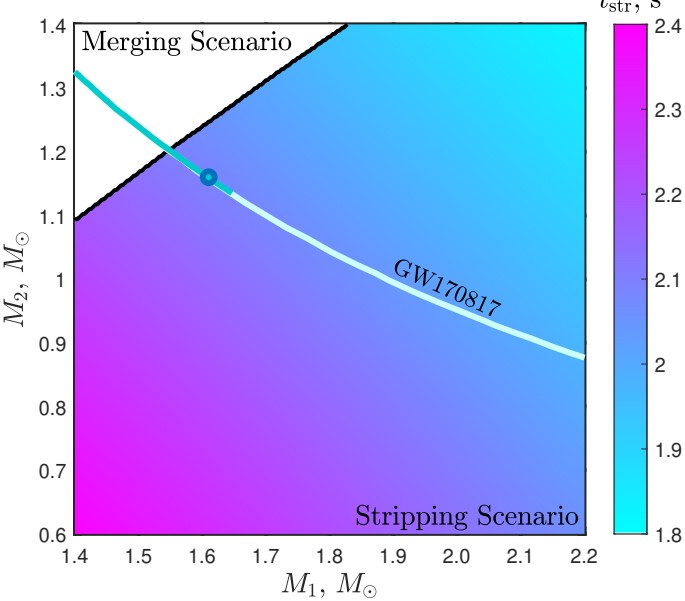

**Figure 3.** The value of $t_{\mathrm{str}}$ for different masses of the binary components for EoS = BSk26. Additionally shown is the boundary between scenarios, the range of masses, dictated by GW170817 and specific values used to build Figure 2 (circled).

Calculations were made for EoS Bsk26 [42]. The mass range corresponding to the GW170817 [43] signal is also shown: the darker part of the curve corresponds to the case of small initial spins of the components, the light part to the large spins. The specific mass value used by us to construct the drawing Figure 2 is shown with a circle. As one can see, there is no value $t_{str} = 1.7$ s for any values of the masses $M_1$, $M_2$, however, this is just a property of the particular EoS that we have used. For other EoSes, this is not the case, but more importantly, as we will show below, there are other factors not taken into account in the above simple version of the stripping mechanism.

Let us also discuss the boundary between scenarios, shown in Figure 3. It is obtained using the following simple considerations: if the overflow occurs, then the low-mass component $M_2$ of the binary system fills its Roche lobe. This means that its radius $R_2$ is equal to the size of the last $R_R$, which [44] may be parameterized well as

$$R_R = af(q),$$

(3)

where, recall, $a$ is the distance between NSs, and $f(q)$ is the well-known [45] function of the mass asymmetry parameter $q = M_2/(M_1+M_2)$. The stability condition for the overflow in this case will then be written as [46]:

$$\frac{d \ln R_2}{d \ln M_2} \geqslant \frac{d \ln f(q)}{d \ln q} - 2\frac{1-2q}{1-q}.$$

(4)

It can be seen that, apart from $q$, the overflow stability is mainly determined by the NS mass-radius relation and, hence, by the equation of state. In the unshaded area in Figure 3, the stable overflow according to the criterion Equation (4) is not possible, and the merging scenario will take place. Leaving until the Section 4 the discussion on what else can affect the parameters of the stripping model and, in particular, the stripping time $t_{str}$, let us now try to answer the following question: why are there still no signs of a matter steady flow in numerous 3D calculations carried out to study the NS merger process?

### 3. Stripping vs Merging: PHANTOM Modelling

Modern 3D calculations of the neutron star merging process are very complex and expensive in terms of computer resources [47]. Therefore, it is extremely important to choose the initial calculation configuration and/or start time correctly. To save resources, there is a temptation to skip the "uninteresting" part: the secular process of approaching of neutron stars (especially since the procedure for correctly accounting for GW radiation in 3D is not at all simple [48]) and to go straight to the more complex dynamical non-linear fusion process. We think that this is at least part of the answer to the question to which this section is devoted.

To conduct numerical experiments in 3D, we chose the open source code PHANTOM [49]. It is based on the method of Smoothed Particle Hydrodynamics, SPH. This is a Lagrangean meshless method for Newtonian dynamics and gravity. Particles in the SPH method are three-dimensional elements of unfixed shape, which have assigned physical characteristics: coordinates, velocity, mass, density, typical size, temperature, pressure, etc. In PHANTOM, a limitation is introduced—all particles have the same mass. Discrete representation of the medium in the form of smoothed particles involves the change of continuous characteristics $f(r)$ into piecewise constants $f_i$, defined for each particle $i$ through the sum $N$ of quantities $f_j$ from the particles of the vicinity $j$, around the particle $i$ using the weight (smoothing) kernel function. Approximation of spatial derivatives in the right-hand parts of equations for conservation laws in SPH is performed through the transfer of derivatives over the particle coordinates to the derivatives of the smoothing kernel function. Based on solutions to the equations of motion, continuity, energy, etc., the particles change position, density, temperature, a new pressure field is calculated for them, and so on. For modeling the close binaries, two equilibrium (relaxed equilibrium star) stars are built in PHANTOM initially, with given masses and radii, with a density profile depending on the equation of state of

matter. The simulation is then started to predict the behavior of such stars already put into orbit.

Consider now the following simplified statement of the problem: take two stars with polytrope EoS $n = 1$ (the adiabatic exponent of the matter is $\gamma = 2$) with masses $M_1 = 1.4 M_\odot$ and $M_2 = 0.5 M_\odot$ and equal radii $R_1 = R_2 = 10$ km. The overflow will begin at the distance between the components of the binary system (see Equation (3)) $a_c = R_2 / f(q) \approx 34$ km. Naturally, one must remember that this expression, derived in the limit of two point bodies, is approximate. Let us set the distance between $a_0$ components to be 30 km, put them in circular Keplerian orbits and run simulations. The result is shown in the Figure 4.

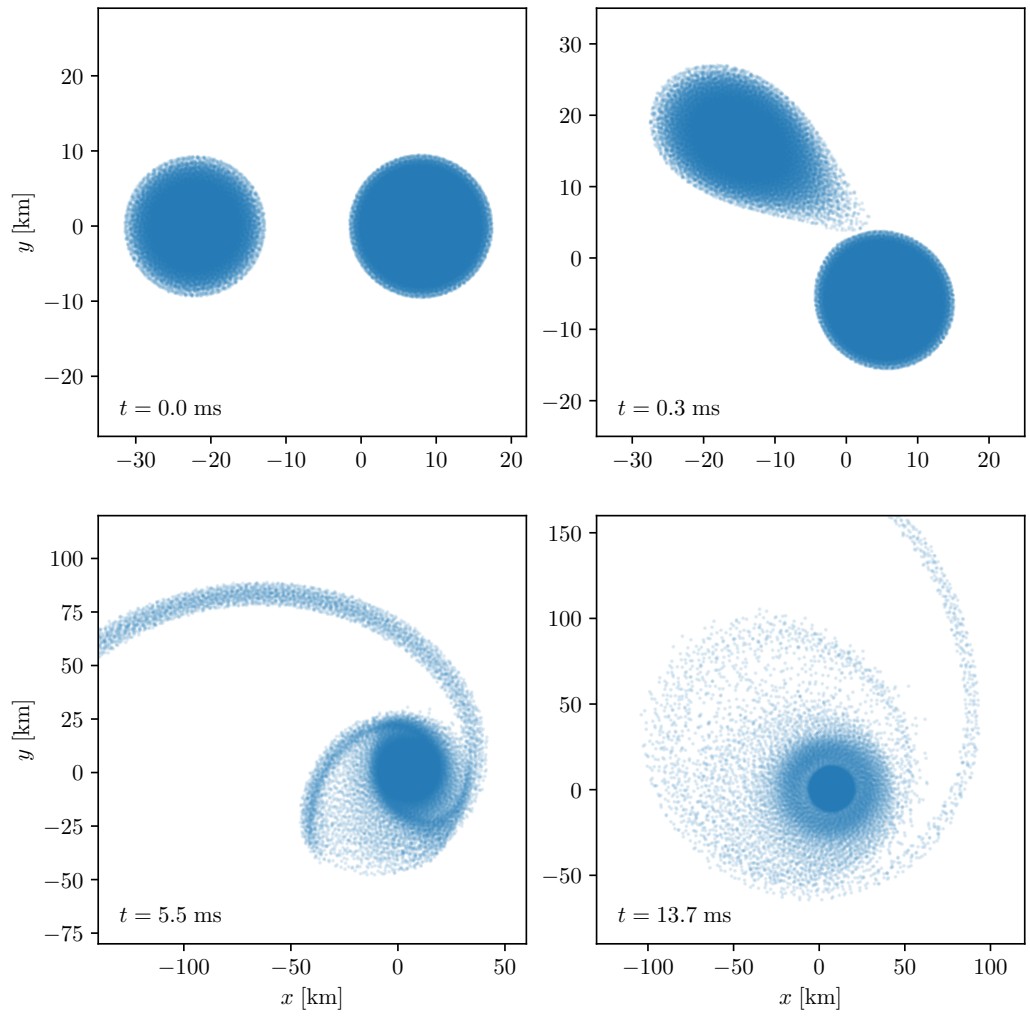

**Figure 4.** Merging binary with two polytropic stars, $M_1 = 1.4 M_\odot$ and $M_2 = 0.5 M_\odot$. The initial distance between the components is 30 km. Time points are indicated on each panel of the figure.

The picture panels represent the projection of what is happening on the $x - y$ plane (the view from the "top") for four moments of time $t$. As one can see, it is a typical merging process, which is preceded by tidal disruption of the low-mass companion.

However, what happens if at the initial moment the components are separated further from each other?

Our numerical experiments have shown that if we start from $a_0 = 36$ km, then the result will be a stable flow of matter (stripping), see Figure 5. It shows only two points in time (left and right panels) corresponding to the beginning and the end of the calculation. As one can see, stripping continues steadily for 20 revolutions (i.e., four times longer than the calculation in Figure 4). We stopped the calculation simply because at such long

time-scales it would be necessary to take into account numerous effects missing from our simple toy model: first of all, the energy losses of the system due to GW radiation and a more realistic equation of state for stars.

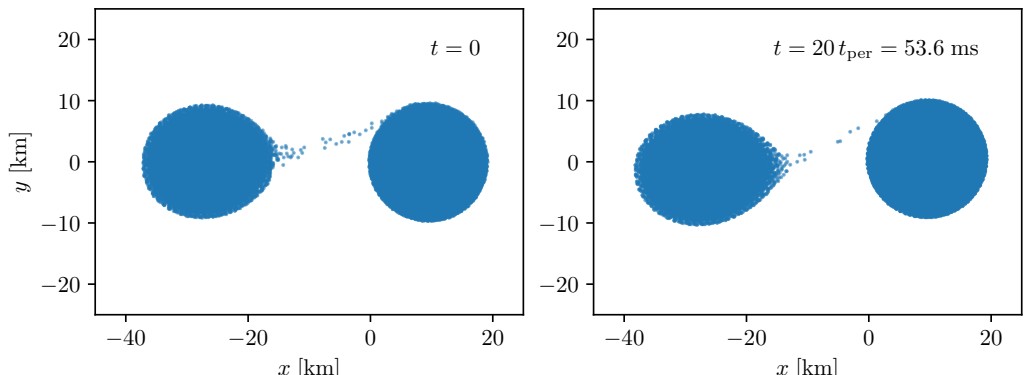

**Figure 5.** Calculation of a binary system evolution with the same parameters as in Figure 4, but with $a_0 = 36$ km.

However, our goal is achieved: we have demonstrated a significant dependence of simulation results on the initial conditions. With a careful selection of the initial conditions and other data the stable stripping regime is obtainable in 3D simulations. In this respect, our conclusions are consistent with those of the important paper [50], where the problem of a correct description of the fate of binary systems of white dwarfs was considered.

In addition, the numerically demonstrated regime of matter stable flow, even in the case of polytropic EoS, does not support the pessimistic conclusion of [28] even in Newtonian regime and inspires a certain confidence. We also numerically investigated cases with other, not so extreme, mass ratios in NS binaries. The existence of a regime of stable mass exchange has also been demonstrated for them. The relevant report is being prepared for publication.

## 4. Discussion

Since the stripping model has received much less attention, than the merging model, many aspects of the former still require further development. We will attempt in this section to list and briefly discuss what remains to be performed to make the predictions of the stripping model on a reliable basis.

The first question, which is common, however, for both models, is the question of the true equation of state for NS. Many predictable parameters of the model indirectly depend on this, in particular, the stripping time $t_{str}$. For our model, the additional complexity is that the region of small NS masses is also important here ($M \lesssim 1 M_\odot$), the description of which, as a rule, does not attract the attention of theorists (see, however, [51]).

The second problem has already been discussed above when describing the simulation in PHANTOM, it is the sensitivity of the results obtained to the initial data. The question actually breaks down into two parts. First, the initial configurations of the stars in any case should be well relaxed [50]. If unperturbed stellar configurations are placed into the orbit at the initial moment of time, this can lead to various phenomena (oscillations and so on), absent in reality. Those phenomena distort the simulation results. The second part of the question concerns the correct account of losses due to GW radiation. As a rule, the influence of losses is taken into account very approximately. For example, in standard PHANTOM setup the gravitational wave radiation reaction acting on each SPH particle within the same star is taken the same [49]. In some models such force is generally neglected [52]. For a highly non-linear merging stage, lasting, as a rule, not more than a few revolutions, such an approach can be justified, however, for the stripping stage this is not at all the case. Actually, it is precisely the losses due to GW radiation that determine the rate of the mass

exchange process. Numerical account of GW-losses in a complex, dynamic binary system of non-point bodies with variable masses is a serious challenge for researchers.

The next important aspect of the mass exchange process in the stripping mechanism was revealed as a result of our numerical experiments using PHANTOM described above. Let us find the relative angular momentum of each star. To do this, we represent the coordinates $\mathbf{r}_a$ and velocities $\mathbf{v}_a$ of SPH-particles with masses $m_a$ confined in the star as:

$$\mathbf{r}_a = \mathbf{r}_a' + \mathbf{R}, \quad \mathbf{v}_a = \mathbf{v}_a' + \mathbf{V}, \tag{5}$$

where

$$\mathbf{R} = \frac{\Sigma m_a \mathbf{r}_a}{\Sigma m_a}, \quad \mathbf{V} = \frac{\Sigma m_a \mathbf{v}_a}{\Sigma m_a}, \tag{6}$$

are the coordinate and velocity of the center of mass of each star. Thus, the hatched variables describe the relative motion of the particles. Then, the relative angular momentum of the star $i$ is equal to:

$$\mathbf{J}_i = \sum_{a \in i} m_a [\mathbf{r}_a' \times \mathbf{v}_a']. \tag{7}$$

It is this value that is presented in the Figure 6 on the left panel, as a function of the calculation time for the stable mass exchange case.

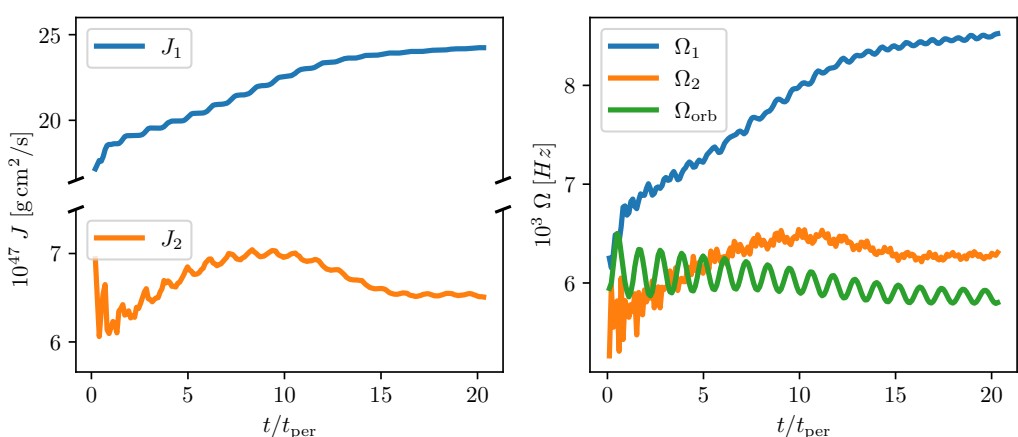

**Figure 6.** Relative angular momentum of two stars (**left panel**) and their angular rotational velocities together with the orbital angular velocity (**right panel**) as functions of time.

On the right panel in the same figure, the effective angular velocities of rotation of both stars are shown, defined as the ratio of their angular momentum to their moments of inertia: $\mathbf{\Omega}_i = \mathbf{J}_i / I_i$. For comparison, the same panel shows the orbital angular velocity of a pair of stars $\mathbf{\Omega}_{\mathrm{orb}}$, found through the rotation velocity of a straight line, linking their centers of mass. Small damped oscillations in $\mathbf{\Omega}_{\mathrm{orb}}$ visible on the graph, are caused precisely by the incomplete relaxation of the initial configuration of the binary system discussed earlier, which is insignificant, however, for this calculation.

As can be seen, the mass exchange in the NS binary system leads to a significant spin-up of a massive companion. Thus, a part of the orbital angular momentum is transferred to the intrinsic spin $M_1$. As our preliminary calculations show, this effect can lead to a serious decrease in $t_{\mathrm{str}}$ stripping time and must be taken into account, in particular, for a correct comparison with observational data (see also Figures 2 and 3).

In conclusion of this section, we formulate a list of questions, still far from complete, requiring their detailed study within the framework of the stripping mechanism. Firstly, this is the fate of the massive NS ($M_1$), whose mass grows as a result of mass exchange. Is it still a fast spinning neutron star accelerating to the critical rotation, due to the spin-up effect discussed above, after which the outflow of matter begins, or does it collapse into a black hole? Additionally, how do these processes affect, in fact, the peeling procedure itself? Related question: what happens to the accreting matter, what is the luminosity

during accretion, whether an accretion disk is formed, and if so, is it possible to form a jet of matter?

The processes occurring in the low-mass component $M_2$ may also turn out to be important. As it fills its Roche lobe, it is subject to tidal heating due to the interaction in the binary system. Its matter experiences a smooth decompression due to a decrease in its density, which can lead to additional heating. LMNS has a specific structure [38]: a small core and an extended shell. Even a relatively weak heating of this shell can greatly affect its structure, and, hence, the rate of mass transfer. In addition, the "hot" structure $M_2$ when it reaches the minimum mass $M_{min}$ may differ from the cold case, which will lead to a difference in the parameters of its explosion, and, therefore, the accompanying GRB.

One can argue that the predictions of a Newtonian code, such as PHANTOM, may not be valid when all effects of GR are properly taken into account, such as in [53] which is based on the hybrid pseudospectral-finite volume Spectral Einstein Code (SpEC). This is one of the longest NS-NS merger simulation about 22 orbits. However, this was an equal mass system with a very idealized EoS. Another important paper in GR is [54] (see also [55]). The most extreme case simulated in the latter paper is a 1:2 mass ratio binary NS simulated with an EoS that predicts radii much larger than currently allowed by GW and X-ray measurements. They had around 10 revolutions and almost no mass transfer prior to merger ($10^{-2} M_\odot$ in their high resolution run). There are tens of other papers citing [53,54], but one should remember that for the case of the stripping scenario the effects of GR may be much weaker than in general merger event, they can well be studied in post-Newtonian approximations. Moreover, some of GR effects (like gravitomagnetism, see, e.g., [56]) must stabilize the process of mass transfer, thus playing in favor of the stable stripping.

Another important aspect still missing in the stripping model is the comparison with optical observations of the kilonova AT∼2017gfo [57]. From general considerations, it is clear that the explosion of the LMNS and accompanying it explosive decompression of approximately $M_{min} \sim 0.1 M_\odot$ highly neutronized matter must be accompanied by significant radiation, powered by the radioactive decay of the elements generated during the explosion. It is extremely interesting to find out whether the stripping scenario has any characteristic fingerprints from merging in this aspect as well. We plan to carry out the corresponding modeling in the near future. This modeling should be facilitated by the fact that, thanks to the geometry of the LMNS explosion, we will be able to carry it out using the author's spherically symmetric STELLA code [58].

## 5. Conclusions

The presented review of the stripping model contains a discussion of many though not all of its aspects. This model experienced a rebirth after the joint discovery of the gravitational wave signal GW170817 and the accompanying weak short gamma-ray burst GRB170817A with the properties predicted already in [26]. Many of the model parameters still need to be refined and tested. A very pressing one, for example, is the question of the boundary of merging–stripping scenarios. The answer to this question will determine the fraction of the stripping mechanism in the general population of short gamma-ray bursts. It should be emphasized here that, in our opinion, both scenarios are not in an antagonistic relationship of the kind "either/or". Rather, they complement each other: under some conditions, the merging takes place, under other circumstance, the stripping does.

The predictions of the stripping mechanism are quite definite: the burst is of low energy, the explosion must be quasi-spherical, etc., [38]. This is an advantage of the stripping over the merging mechanism, where there are many "configuration" options: like varying the direction of the jet, its opening angle [59], the structure [60], parameters of the surrounding matter (cocoon, "choked" jet [61]), and so on. In this way one can, in fact, explain any observational data based on merging scenario. The stripping model is devoid of such arbitrariness. On the other hand, of course, there is a significant number of common uncertainties in both mechanisms, such as the unknown equation of state of superdense matter,

difficulties, and inevitable simplifications in 3D modeling of hydrodynamic phenomena in GR, and so on.

Even if the fraction of the stripping mechanism in the total GRB population is small, its influence in some respects can be significant: it should be remembered that the mass of the ejected matter here is equal by the order of magnitude to the minimum NS mass $M_{\min} \sim 0.1 M_\odot$, which is a lot, see, for example, [57]. Preliminary calculations of nucleosynthesis [62] accompanying the $M_2$ explosion look very promising, which, coupled with the large amount of ejected mass, allows us to hope for a large contribution of the stripping mechanism to the cosmic process of formation of heavy elements.

**Author Contributions:** Conceptualization, S.B. and A.Y.; writing, A.Y.; review and editing, S.B., N.K. and M.P.; calculations with PHANTOM, M.P. and S.B., stripping calculations, N.K. All authors have read and agreed to the published version of the manuscript.

**Funding:** Authors are grateful to RFBR grant 18-29-21019mk for support. S. Blinnikov is partly supported by RFBR 21-52-12032 grant. PHANTOM modeling by M. Potashov is supported by RSCF grant 19-12-0229.

**Data Availability Statement:** Not Applicable.

**Acknowledgments:** We dedicate this work to the memory of D.K. Nadyozhin, our friend and teacher, who made an invaluable contribution both to astrophysics in general and to the development of the stripping scenario in particular. Authors are grateful to referees for their valuable comments. We also thank Anastasia Buinovskaya for colorful and informative illustration. S.Blinnikov is very grateful to Gevorg Hajyan for the warm hospitality and to Olga Sheetova for her valuable assistance during his visit to Yerevan.

**Conflicts of Interest:** The authors declare no conflict of interest.

## Abbreviations

The following abbreviations are used in this manuscript:

| | |
|---|---|
| EoS | Equation of State |
| GRB | Gamma-Ray Burst |
| GW | Gravitational Wave |
| LMNS | Low-Mass NS |
| NS | Neutron Star |
| SN | Supernova |
| SPH | Smoothed Particle Hydrodynamics |
| VLBI | Very Long Baseline Interferometry |

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
