# Peer review of "Stripping Model for Short Gamma-Ray Bursts in Neutron Star Mergers"

_2571-712X, doi:10.3390/particles5020018_

Round 1

Reviewer 1 Report

Dear Authors,

I read carefully the manuscript "Stripping model for short gamma-ray bursts in neutron star mergers" submitted to Particles. The draft presents a brief description of a scenario where the gamma-ray burst detected in association with the 

1. Introduction:
- l. 11: neither a reference nor a definition is given for "collapsing hypernovae", which seems jargon from a particular niche within the gamma-ray burst community. Please clarify.
- l. 24: "In this case, part of the matter ...". In sufficiently asymmetric binaries, tidal interactions can lead to matter ejection even in cases where the merger results in a prompt collapse to a BH, see e.g. Bernuzzi et al. 2020.
- l. 25: "[...] due to the extreme acceleration of the rotation of matter, [...] collimated ejection of matter can form ...". This description of the process that leads to jet launching is unclear, mostly incorrect, and lacks a refrence. As a guidance, let me summarize the status of this complex and highly debated problem, as far as my knowledge goes: the main jet launching mechanism candidates are, in presence of a black hole remnant and a non-negiglible accretion disk, the Blandford-Znajek process (Blandford & Znajek 1977, see e.g. Christie et al. 2019 for a recent discussion of the role of magnetic field geometry) and the neutrino-antineutrino annihilation process (Eichler et al. 1989, but see e.g. Just et al. 2016 for results that disfavour this as the main jet-powering process in binary neutron star mergers). Alternatively, jet launching might be possible also in the case the remnant is a rapidly rotating, highly magnetized proto-neutron star (Usov 1992, C. Thompson 1994, T. Thompson 2004, Metzger et al. 2011, but how this is realistic is uncertain, see e.g. Ciolfi 2020 and Moesta 2020 for recent contrasting results).
- l. 50: "... the stripping scenario was forgotten for many years ...". I think a fair historical reconstruction should mention the result by Lai, Rasio & Shapiro 1994 that mass transfer in NS-NS binaries is very hardly stable (the authors may possibly want to look for some more recent reference).

2. GRB170817A and the stripping model

- l. 64: "... showed no signs of a strong jet". Letting aside the expression "strong jet" being obscure, this statement is not up to date: subsequent observations clearly showed strong signatures of a narrow relativistic jet observed slightly off-axis (Mooley et al. 2018, Nature, 561,  7723; Mooley et al. 2018, ApJL, 868, 1, L11; Ghirlanda et al. 2019, Science, 363, 6430).
- figure 2: it is unclear how the evolution in fig. 2 is computed, but it seems to imply an anti-chirp GW signal that is not present in the GW170817 data, despite the values of a (and hence the GW frequency, since the total mass remains constant) and L_GW being similar for at least half a second before and after the L_GW peak.
- l. 82: the fact that the putative low-mass NS explosion in the model happens ~2 seconds after the GW peak could be an issue itself: the ejecta will still need to expand enough as to reach their Compton photosphere before being able to produce gamma rays. If the mass of the gamma-ray producing shell is a fraction f of the exploding NS mass (M_exp ~ 0.1 Msun), then the Compton photosphere is approximately R_ph ~ (sigma_T f M_exp / 4 pi m_p)^1/2 , where sigma_T is the Thomson cross section and m_p is the proton mass. This is R_ph ~ 8x10^14 cm (f/0.1)^0.5, which leads to a time to reach the photosphere, in the observer frame, of t_ph ~ G^-2 R_ph/c ~ 270 s (G/10)^-2 (f/0.1)^0.5, where G is the shell's Lorentz factor. The gamma-ray producing shell must therefore either carry a tiny fraction of the M_exp (f<10^-5) or be ultrarelativistic (G>100). It is unclear to me whether any of these conditions can be realistically achieved (according to Colpi, Shapiro and Teukolsky 1993, the typical ejecta are sub-relativistic (by the way, they also find that the explosion onset is far from instantaneous).

3. Stripping vs merging: PHANTOM modelling and 4. Discussion

Honestly, I do not see the whole point of section 3. The calculations presented here are of little use, given that they represent at best a toy model. The sensitivity on the way initial conditions are generated for Newtonian SPH simulations of polytropic binaries has been already investigated by Dan et al. 2011, who showed the need for such initial data to be produced carefully by relaxation schemes, rather than made "by hand". For what concerns the issue of resolving mass transfer in general-relativistic simulations with the relevant physics, I think the couple of references mentioned in sec. 4, and the discussion around them, are insufficient. Perhaps a more relevant result is that from Dietrich et al. 2015 (https://ui.adsabs.harvard.edu/abs/2015PhRvD..92l4007D/abstract), but this is definitely showing a different behaviour with respect to that described in the present manuscript.

A point which is missing in the discussion is how the at2017gfo kilonova observations would fit in the stripping scenario, given how detailed the data are and how generally good the agreement is with the "standard" scenario.

5. Conclusions

I find the point that "the stripping scenario is more definite and less uncertain than the mergin scenario" in clear contradiction with the arguments presented earlier in the draft, where the authors state that it has been given much less attention and studied less: the uncertainties of the "merging scenario" come precisely from the fact that it has been studied in detail and the high complexity of the processes involved has been explored to some extent. This is clearly to be expected in the stripping scenario as well. 

Author Response

  1. Q. neither a reference nor a definition is given for "collapsing hypernovae"
    A. Several references and explanations are given on GRB-HN connection, where HN is a hypernova produced in a collapse of a stripped massive star.
  2. Q. In sufficiently asymmetric binaries, tidal interactions can lead to matter ejection…
    A.
    Paragraph on tidal interactions reformulated and fixed, Bernuzzi et al. 2020 citation added.
  3. Q. This description of the process that leads to jet launching is unclear mostly incorrect, and lacks a reference.
    A. We are grateful to the referee for detailed discussion on the problems of jet formation, although those details are outside of the main focus of our paper. We added almost all references cited by the referee up to Ciolfi 2020 and Moesta 2020 for recent contrasting results on jets and we have added a few historical references on jet formation which seem important to us from our perspective.

  4. Q. a fair historical reconstruction should mention the result by Lai, Rasio & Shapiro 1994... 
    A. An important reference to Lai, Rasio & Shapiro 1994 is added into introduction and to the discussion of SPH simulations.

  5. Q. Letting aside the expression "strong jet" being obscure, this statement is not up to date…
    A. Reformulated, relevant references are added.

  6. Q.  it is unclear how the evolution in fig. 2 is computed. 
    A. The evolution of the NS-NS system in Fig. 2 is calculated in a way similar to Jaranowski and Krolak, ApJ, 1992. The signal is asymmetric, but close to symmetry in a diapason from -0.2 to +0.2 sec approx. The signal after the L_GW peak was not identified in the GW170817 because of the absence of a corresponding waveform model (see, for example, Mishra et al, PhRD, 2016).

  7. Q.  the fact that the putative low-mass NS explosion in the model happens ~2 seconds after the GW peak could be an issue itself…
    A.
    This is a really important question. The answer to it is related to how gamma radiation is formed in the stripping scenario. The whole process of a neutron star explosion takes a fraction of a second. At the same time, only a very small thin region near the surface of the star is heated to temperatures greater than or on the order of about 1GK: see Fig. 12 from the paper by Blinnikov et al. 1990. It is necessary to pay attention to the logarithmic scale on the x axis and the fact that the mass dM here is calculated from the surface of the star. The same thin layer, small in mass, accelerates to semi-relativistic velocities. The process of the shock breakout through the surface, its accumulation and heating are also shown in the attached file from the upcoming article. The coordinate m is in the masses of the Sun, the whole process takes about 7 ms. The corresponding changes have been made to the text.

  8. Q. Honestly, I do not see the whole point of section 3…
    A.
    The presented calculations, although with simplified physics, are intended to illustrate several ideas: a) the sensitivity of the simulation results to input parameters, including the final fate of the merging/stripping system b) the existence of a stable flow regime, on which, as is correctly pointed out by the referee, there were and still are doubts and c) the existence and importance of the spin-up effect. What is the reason for the discrepancy with the conclusions of Dietrich et al. - it remains to be seen, this is our ultimate goal, we are currently working on preparing for the implementation of the relevant calculations. Let us not forget that the calculations we have presented are just an illustration of the stripping mechanism  in our mini-review, and not a great achievement in itself. Relevant references have been added. 

  9. Q. A point which is missing in the discussion is how the at2017gfo kilonova observations would fit in the stripping scenario…
    A.
    The words about at2017gfo have been added to the discussion. We plan to make such a calculation using our STELLA code. It is worth noting here that in the standard scenario there are significant uncertainties and problems with explaining, for example, the properties of the so-called red kilonova (Siegel 2019) which is quite naturally explained (actually, predicted) in the stripping scenario.

  10. Q. This [uncertainty] is clearly to be expected in the stripping scenario as well.
    A.
     Reformulated. We just wanted to emphasize here that in addition to the uncertainties in input physics inherent in both scenarios, the merging scenario is also accompanied by uncertainties due to unknown jet parameters, etc., which may vary from system to system. And in stripping there are no such “configurable” parameters - the neutron star of the minimum mass always explodes. In this regard, stripping's GRB should be something like a “standard candle". The situation here is somewhat similar to the comparison between core-collapse SN (huge variability in the light curves) and SN Ia - standard candles.

    Our goal in the current mini-review is not to solve all the problems of the stripping scenario but to draw the attention of the community to this model which has many promising advantages.

Reviewer 2 Report

In this work the authors have reviewed the stripping model for the mergers of binary neutron stars with different masses, which eventually lead to short gamma-ray bursts as matter being stripped from the low mass neutron star to the more massive one. The stripping time can then explain the 1.7 second delay between the trigger times of the gravitational-wave signal GW170817 and the γ-ray burst GRB 170817A. In my opinion, the stripping model should be viewed as a special scenario for binary neutron star mergers. I can recommend its publication after the authors consider my remarks and introduce more discussions, i.e.,

  1. The authors should clarify the final stage where the low mass neutron star reaches its minimum value (~0.1 M⊙). The radius would increase significantly if the neutron star is isolated and static. However, this is not the case since a more massive neutron star is still interacting with it. Does it end up like the merger scenario of a binary system with large mass asymmetries, or becomes a combination of a neutron starand a white dwarf?
  2. As large amount of matter is ejected according to the stripping model(as indicated in line 245), the ejecta could be optically thick to γ-rays and further delay the γ-ray bursts. The authors should add some comments on this.

Author Response

  1. Q. Does it end up like the merger scenario of a binary system with large mass asymmetries, or becomes a combination of a neutron star and a white dwarf?
    A. A neutron star of a smaller mass, indeed, inflates in the process of stripping, but not to the size of WD, but to about 300 km, where the loss of stability and explosion occurs. If we turn to our Fig.3, it can be seen that as a result of the evolution of the system (M1 is growing, M2 is falling) we are moving downwards to the right, i.e. away from the merging scenario. In addition, as can be seen from Fig. 2, the distance between the companions is increasing more and more. The really important issue raised by the referee here is the influence of a massive companion. In Manukovskii 2010, the influence of a massive companion was studied and it was shown that even in its presence the explosion quickly symmetrizes (see Fig. 7 from this article), and the full parameters of the explosion are close to the spherically symmetric case.

  2. Q. As large amount of matter is ejected according to the stripping model (as indicated in line 245), the ejecta could be optically thick to γ-rays and further delay the γ-ray bursts. The authors should add some comments on this.
    A. This is a really important question. The answer to it is related to how gamma radiation is formed in the stripping scenario. The whole process of a neutron star explosion takes a fraction of a second. At the same time, only a very small thin region near the surface of the star is heated to temperatures greater than or on the order of about 1GK: see Fig. 12 from the work of Blinnikov et al. 1990. It is necessary to pay attention to the logarithmic scale on the x axis and the fact that the mass dM here is calculated from the surface of the star. The same thin layer, small in mass, accelerates to semi-relativistic velocities. The process of the shock breakout through the surface, its accumulation and heating are also shown in the attached file from the upcoming article. The coordinate m is in the masses of the Sun, the whole process takes about 7 ms. The corresponding changes have been made to the text.

Reviewer 3 Report

The present article explains an alternative model to the standard neutron star merger scenario, namely the stripping model. The article provides a basic introduction to the model and demonstrates several numerical experiments within this model. The article is timely and provides an excellent introduction to this fascinating topic. I recommend the acceptance of the article.

Author Response

We are grateful to the referee for his encouraging report.

Round 2

Reviewer 1 Report

Dear Authors,
I read the revised manuscript, and I find that most of the issues from my previous report are still standing, and new ones have been raised by the new text. These prevent me from recommending the publication of the manuscript in its present form. The authors do not seem to be willing to take my concerns seriously, therefore I do not see good prospects for getting to a publication. Here are the main issues:

- l. 83: in the newly added text, the authors are presenting a distorted reconstruction of the interpretation of GW170817. It is unclear to me whether this is on purpose (in an attempt to hide the overwhelming evidence towards the presence of a highly collimated relativistic jet in that source) or it is due to a lack of understanding. In any case, this part should be amended. Ref. [34] predates ref. [33], as can be simply checked by looking at the submission dates on arXiv. Therefore, saying that the authors of [34] criticize [33] is a distortion or, at best, nonsense. The superluminal motion found in [33] (and confirmed in Ghirlanda et al. 2019, Science) is impossible to accommodate in a spherical explosion scenario (it requires a highly anisotropic, relativistic expansion, i.e. a relativistic jet). On top of this, the radio and X-ray light curve decay after the peak is too steep for a spherical explosion (Mooley et al. 2018, ApJL, 868, 1, L11; Lamb, Mandel & Resmi 2018, 481, 2, 2581) - which also contradicts the last statement in the new text. It is unclear, moreover, how the Hajela et al. reference fits here, since that work is about a late-time X-ray excess *on top of the jet afterglow*. Please fix this text and simply recognize that the presence of a relativistic jet in GW170817 is supported by strong evidence. As a final note, the fact that the prompt emission was 10^4 times fainter than typical SGRBs is because of the viewing angle, which strongly suppresses the jet core emission, as seen from Earth, due to relativistic beaming of radiation: indeed, the GRB170817A gamma-rays were likely not produced in the jet core (as were presumably those seen in other gamma-ray bursts) but several degrees away from it (either by internal dissipation, or by the cocoon shock breakout).

- Figure 2: in the reply to my first report, the authors state that "The evolution of the NS-NS system in Fig. 2 is calculated in a way similar to Jaranowski and Krolak, ApJ, 1992": this should be added to the text.  The authors also state that "The signal after the L_GW peak was not identified in the GW170817 because of the absence of a corresponding waveform model (see, for example, Mishra et al, PhRD, 2016)": I could not find the cited reference, but the authors seem to mention the fact that GW searches are performed using matched filtering techniques based on pre-computed signals. This is only partly true: while the most sensitive pipelines such as GSTLaL and pyCBC do use that approach, the data are also analyzed with agnostic pipelines such as X-pipeline and Coherent Wave Burst. Moreover, the kind of signal that would be produced according to the evolution shown in Fig. 2 would be visible *by eye* in the time-frequency diagrams (Fig. 1 of the GW170817 discovery paper, Abbott et al. 2017, PRL, 119, 161101).

- l. 180: the authors state that they have "numerically demonstrated" a regime of stable flow of matter, which is an unacceptable statement, given the inadequacy of the physics description in their simulation, and the fact that much more advanced simulations yield exactly the opposite answer.

- "Our goal in the current mini-review is not to solve all the problems of the stripping scenario but to draw the attention of the community to this model which has many promising advantages". I am not asking the authors to solve the problems of the stripping scenario. My role as a referee is that of ascertaining that the methods, statements and conclusions of this work are neither incorrect, nor misleading. Unfortunately, at the present stage, these requirements are not satisfied, and therefore I cannot recommend the publication of the manuscript on any serious journal.

Author Response

>I read the revised manuscript, and I find that most of the issues from my previous report are still standing, and new ones have been raised by the new text. These prevent me from recommending the publication of the manuscript in its present form. The authors do not seem to be willing to take my concerns seriously, therefore I do not see good prospects for getting to a publication.

A.  From about 10 of his comments only 3 are still bothering the referee, so this remark is not fare.

> - l. 83: in the newly added text, the authors are presenting a distorted reconstruction of the interpretation of GW170817. It is unclear to me whether this is on purpose (in an attempt to hide the overwhelming evidence towards the presence of a highly collimated relativistic jet in that source) or it is due to a lack of understanding. In any case, this part should be amended. Ref. [34] predates ref. [33], as can be simply checked by looking at the submission dates on arXiv. Therefore, saying that the authors of [34] criticize [33] is a distortion or, at best, nonsense.

A. Now we give this paragraph in the following redaction:

Another model, that of the choked jet, was advanced in~\cite{Nakar2018} by some authors of~\cite{Mooley2018}.
Although, \cite{Nakar2018} predates  \cite{Mooley2018}, the citation and discussion of ~\cite{Mooley2018} is in footnote~6
of~\cite{Nakar2018}, where the latter write that ``choked jet seems to be incompatible with GW1708178''.
Nevertheless, X-ray observations do not exclude that afterglow emission arises from
  a quasi-spherical mildly relativistic outflow~\cite{Nynka2018,Hajela2022}.
Moreover,  recent data confirm ``a growing tension between the observations and the jet model''~\cite{Troja2022}.
However, as we have shown in \cite{BlinnikovNadyozhinKramarevEtal2021},
  many features of GRB170817A get their natural explanation in the short GRB stripping mechanism.

>The superluminal motion found in [33] (and confirmed in Ghirlanda et al. 2019, Science) is impossible to accommodate in a spherical explosion scenario (it requires a highly anisotropic, relativistic expansion, i.e. a relativistic jet).

A. In general this statement is not true. An apparent superluminal motion is observed when a cool perfectly spherical semi-relativistic shell shocks a cloud positioned
at the same angle towards the observer which people derive in their jet model. But NO JET is involved in this process!

> - Figure 2: in the reply to my first report, the authors state that "The evolution of the NS-NS system in Fig. 2 is calculated in a way similar to Jaranowski and Krolak, ApJ, 1992": this should be added to the text.

A. This was already there in the text in the previous revision.

>The authors also state that "The signal after the L_GW peak was not identified in the GW170817 because of the absence of a corresponding waveform model (see, for example, Mishra et al, PhRD, 2016)": I could not find the cited reference, but the authors seem to mention the fact that GW searches are performed using matched filtering techniques based on pre-computed signals. This is only partly true: while the most sensitive pipelines such as GSTLaL and pyCBC do use that approach, the data are also analyzed with agnostic pipelines such as X-pipeline and Coherent Wave Burst. Moreover, the kind of signal that would be produced according to the evolution shown in Fig. 2 would be visible *by eye* in the time-frequency diagrams (Fig. 1 of the GW170817 discovery paper, Abbott et al. 2017, PRL, 119, 161101).

A. Let us check with the paper Abbott_2017_ApJL_848_L12.

The Astrophysical Journal Letters, 848:L12 (59pp), 2017 October 20
https://doi.org/10.3847/2041-8213/aa91c9

Title: Multi-messenger Observations of a Binary Neutron Star Merger

Authors: LIGO Scientific Collaboration and Virgo Collaboration, Fermi
GBM, INTEGRAL, IceCube Collaboration, AstroSat Cadmium Zinc...

They wrote:

2.1. Gravitational-wave Observation
GW170817 was first detected online (Cannon et al. 2012;
Messick et al. 2017) as a single-detector trigger and disseminated
through a GCN Notice at 13:08:16 UTC and a GCN Circular at
13:21:42 UTC (LIGO Scientific Collaboration & Virgo Collabora-
tion et al. 2017a). A rapid re-analysis (Nitz et al. 2017a, 2017b) of
data from LIGO-Hanford, LIGO-Livingston, and Virgo confirmed
a highly significant, coincident signal.

-- end citation of ApJL --

Let us look into the cited telegram:

LIGO Scientific Collaboration & Virgo Collaboration et al. 2017a:

TITLE:   GCN CIRCULAR
NUMBER:  21505
SUBJECT: LIGO/Virgo G298048: Fermi GBM trigger 524666471/170817529:
LIGO/Virgo Identification of a possible  gravitational-wave
counterpart
DATE:    17/08/17 13:21:42 GMT
FROM:    Reed Clasey Essick at MIT  <[email protected]>

The LIGO Scientific Collaboration and the Virgo Collaboration report:

The online CBC pipeline (gstlal) has made a preliminary
identification of a GW candidate associated with the time
 of Fermi GBM trigger 524666471/170817529 at gps time 1187008884.47
 (Thu Aug 17 12:41:06 GMT 2017) with RA=186.62deg Dec=-48.84deg and an
error radius of 17.45deg.

The candidate is consistent with a neutron star binary coalescence with
False Alarm Rate of ~1/10,000 years.

An offline analysis is ongoing. Any significant updates will be provided
by a new Circular.

[GCN OPS NOTE(17aug17): Per author's request, the LIGO/VIRGO ID
was added to the beginning of the Subject-line.]

-- end citation of GCN CIRCULAR --

CBC pipeline (gstlal) is described in detail here:

https://emfollow.docs.ligo.org/userguide/analysis/searches.html

Online Pipelines
A number of search pipelines run in a low latency, online mode. These
can be divided into two groups, modeled and unmodeled. The modeled
(CBC) searches specifically look for signals from compact binary
mergers of neutron stars and black holes (BNS, NSBH, and BBH systems).
The unmodeled (Burst) searches on the other hand, are capable of
detecting signals from a wide variety of astrophysical sources in
addition to compact binary mergers: core-collapse of massive stars,
magnetar star-quakes, and more speculative sources such as
intersecting cosmic strings or as-yet unknown GW sources.

Modeled Search
GstLAL, MBTA, PyCBC Live and SPIIR are matched-filtering based
analysis pipelines that rapidly identify compact binary merger events,
with ≲1 minute latencies. They use discrete banks of waveform

templates to cover the target parameter space of compact binaries,
with all pipelines covering the mass ranges corresponding to BNS,
NSBH, and BBH systems.

-- end citation of userguide --

GstLAL -- "they use discrete banks of waveform templates"

So, even the alert signal is made according to templates, not to mention Fig.1,
pointed out by the referee. For the offline processing, everything is done according to detailed templates (as describe in Sec.III of the paper cited by referee, Abbott et al. 2017, PRL, 119, 161101). The templates are
based on post-Newtonian calculations, which are not suitable for the violent merging phase at all.

No one saw anything by eye, and if they had, then at the beginning of violent merging, the signal would only have intensified, and not disappeared.
In our model, it is on the decline. If the referee believes that the signal is visible *by eye*
he may try to see this declining behavior in the attached time-frequency diagram
(encircled by red line).
